

# Identification of candidate reference genes for qRT-PCR normalization studies of salinity stress and injury in *Onchidium reevesii*

Teizhu Yang[1,2,3,*], Bingning Gu[1,2,3,*], Guolyu Xu[1,2,3], Yanmei Shi[1,2,3], Heding Shen[1,2,3], Rongcheng Rao[1,2,3] and Hellen Lucas Mzuka[1,2,3]

[1] National Demonstration Center for Experimental Fisheries Science Education, Shanghai, China
[2] Key Laboratory of Exploration and Utilization of Aquatic Genetic Resources (Shanghai Ocean University), Ministry of Education, Shanghai, China
[3] Shanghai Universities Key Laboratory of Marine Animal Taxonomy and Evolution, Shanghai, China
[*] These authors contributed equally to this work.

## ABSTRACT

Real-time quantitative reverse transcription-PCR (qRT-PCR) is an undeniably effective tool for measuring levels of gene expression, but the accuracy and reliability of the statistical data obtained depend mainly on the basal expression of selected housekeeping genes in many samples. To date, there have been few analyses of stable housekeeping genes in *Onchidium reevesii* under salinity stress and injury. In this study, the gene expression stabilities of seven commonly used housekeeping genes, *CYC*, *RPL28S*, *ACTB*, *TUBB*, *EF1a*, *Ubiq* and *18S RNA*, were investigated using BestKeeper, geNorm, NormFinder and RefFinfer. Although the results of the four programs varied to some extent, in general, *RPL28S*, *TUBB*, *ACTB* and *EF1a* were ranked highly. *ACTB* and *TUBB* were found to be the most stable housekeeping genes under salinity stress, and *EF1a* plus *TUBB* was the most stable combination under injury stress. When analysing target gene expression in different tissues, *RPL28S* or *EF1a* should be selected as the reference gene according to the level of target gene expression. Under extreme environmental stress (salinity) conditions, *ACTB* (0 ppt, 5 ppt, 15 ppt, 25 ppt) and *TUBB* (35 ppt) are reasonable reference gene choices when expression stability and abundance are considered. Under conditions of 15 ppt salinity and injury stress, our results showed that the best two-gene combination was *TUBB* plus *EF1a*. Therefore, we suggest that *RPL28S*, *ACTB* and *TUBB* are suitable reference genes for evaluating mRNA transcript levels. Based on candidate gene expression analysis, the tolerance of *O. reevesii* to low salinity (low osmotic pressure) is reduced compared to its tolerance to high salinity (high osmotic pressure). These findings will help researchers obtain accurate results in future quantitative gene expression analyses of *O. reevesii* under other stress conditions.

Corresponding author
Heding Shen, hdshen@shou.edu.cn

## INTRODUCTION

With advantages of relatively accurate quantification, high sensitivity and high throughput, quantitative reverse transcription polymerase chain reaction (qRT-PCR) has become one of the most widely used techniques to detect changes in gene expression (*Johnson et al., 2014*; *Phillips et al., 2009*). Indeed, the use of qRT-PCR has increased tremendously in nearly all branches of biology (*Jian et al., 2008*; *Yang et al., 2018*; *Leidinger et al., 2016*; *Chervoneva et al., 2017*). However, there are inevitably a number of influencing factors that affect the efficiency of the reaction (*Bustin et al., 2009*), such as discrepancy in pipetting, the RNA quality and concentration, the efficiency of reverse transcription and amplification among different samples, the PCR procedures and primer amplification efficiency. Therefore, to avoid variations or errors, it is fundamental to standardize the level of target gene expression by utilizing in parallel a reference gene as an internal control (*Huggett et al., 2005*). In general, an ideal reference gene should demonstrate a consistent level of expression across all tested tissues or conditions (*Bustin & Bustin, 2002*). Nonetheless, there is increasing evidence that expression of assumed reference genes can vary observably with experimental conditions such as developmental stage and chemical treatment, significantly affecting relative quantification and qPCR result interpretation.

Certain housekeeping genes, such as *18S ribosomal RNA* (*18SrRNA*), *28S ribosomal RNA* (*28SrRNA*), *β-actin* (*ACTB*), *cyclophilin* (*CYP2*), *elongation factor 1-alpha* (*EF1a*), *glyceraldehyde-3-phosphate dehydrogenase* (*GAPDH*), *translation elongation factor* (*TEF*), *tubulin* (*TUBB)* and *polyubiquitin* (*UBQ*) (*Huan, Wang & Liu, 2017*; *Song et al., 2017*; *Gao et al., 2017*) are commonly used as reference genes. As these housekeeping genes are related to basic metabolic processes and are essential for normal cell growth, their expression levels are thought to be stable. However, many recent studies have revealed that fluctuations in the expression levels of housekeeping genes may be largely influenced by the tissue, individual developmental stage or experimental conditions (*Hu, Xie & Yao, 2016*). In fact, there is no clear evidence to show that a single universal reference gene is suitable for different tissues and varying experimental conditions (*Vandesompele et al., 2002*). Thus, it is crucial to determine one or two stably expressed reference genes before they are applied for normalizing the expression levels of target genes based on qRT-PCR.

*Onchidium reevesii* (Mollusca, Gastropoda, Pulmonata, Systellommatophora, Onchidiidae) is a brackish water amphibious sea slug that resides mainly in river ports of the intertidal zone and coastal tidal flats, reed and mangroves ecosystems (*Wang et al., 2018*; *Sun et al., 2014*). The species is rich in natural collagen, which has high nutritional and medicinal value; thus, it may constitute an excellent specialty aquatic product for humans if issues pertaining to their breeding in captivity can be resolved (*Guan et al., 2013*). Moreover, *O. reevesii* is an important part of the biodiversity of wetlands and is considered to be a model organisms for studying the evolution of marine invertebrates from sea to land (*Sun et al., 2016*). *O. reevesii* has been investigated with regard to morphological, physiological and active substance aspects, and further molecular biology research is urgent (*Wang, 2014*; *Shen et al., 2010*; *Cheng et al., 2015*). Additionally, suitable reference genes are important for accurately interpreting expression analyses of functional genes and for

evaluating RNAi efficiency. *O. reevesii* is a euryhaline organism, and the salinity of its habitat varies greatly under the influence of tides, rainfall, river flow and other factors. Due to its complex geographical environment and biodiversity, these organisms are inevitably attacked and injured by natural enemies. In this study, an extreme environment (salinity) and stress conditions were imposed on *O. reevesii* in an effort to determine stably expressed internal reference genes under such treatment and to provide basic data for future studies on its ability to regulate osmotic pressure, to adapt to extreme environments and to repair damage.

## MATERIAL AND METHODS

### Animals and treatments

Shanghai Ocean University of Leicester granted Ethical approval to carry out the study within its facilities (Approval number: Shou-DW-2019-010). Sexually mature *O. reevesii* adults were obtained from the coast of Shanghai, China, and housed in the laboratory according to Shen's method (*Shen et al., 2004*) in aquaculture tanks (each tank containing no more than 50 individuals), with enough fine silt and seawater (10 ppt) to simulate their natural living environment and shelters for the animals to hide. Feeding (corn flour and diatoms), removal of faeces and fresh seawater replacement were performed in a timely manner. The animals were allowed to acclimate to the new environment for 7 days before the experiments. For salinity stress, 150 *O. reevesii* individuals of the same size and weight were divided into five salinity groups (0 ppt, 5 ppt, 15 ppt, 25 ppt, 35 ppt). The water used in the experiment was saline, and the volume of water used ensured that all samples were retained. For injury stress, a surgical blade was used to inflict wounds of ~5–7 mm long and 2–3 mm deep on the dorsal skin of 30 *O. reevesii* individuals; the experiment was divided into 5 groups according to the weight of the individuals: 8 g, 11 g, 15 g, 18 g and 22 g. To eliminate the influence of temperature change, the experimental temperature was maintained at approximately 25 °C, the temperature at which *O. reevesii* exhibits the best activity in the field.

### Sampling

Dorsal skin tissues were used as samples in salinity stress and injury experiments; sampling was performed at 2 h, 4 h, 12 h, 24 h, 48 h and 7 days. Three individuals from each group were sampled at every time point (*Bai et al., 2014*). Dorsal skin tissues were also utilized as samples for the groups of different weights. Six tissues were used for assessing gene expression, including the dorsal skin muscle, intestine, lip, pleopod, liver pancreas and gonad. All of the samples were placed in freezer tubes, frozen in liquid nitrogen and stored at −80 °C.

### Total RNA extraction and first-strand cDNA synthesis

Samples were rapidly ground in liquid nitrogen, and total RNA was extracted using TRIZOL (TaKaRa, Otsu, Japan). All RNA samples were resuspended in RNase and DNase-free ddH$_2$O (TaKaRa, Otsu, Japan). The integrity of total RNA was determined by 1% agarose gel electrophoresis, and a NanoDrop 2000c spectrophotometer (Thermo Scientific,

**Table 1  Details of the primers used for qRT-PCR normalization.**

| Gene abbreviation | Gene name | Biological function | Primer sequence (5′–3′) | Product size (bp) | Accession no. |
|---|---|---|---|---|---|
| CYC | cyclophilin | Immunosuppressant, protein folding | F: GTGGGATGTTCCTCTTTACC<br>R: ACCTGGGATTATTCTGTGG | 269 | KY593122 |
| RPL28 | Ribosomal protein l 28 | 60S ribosomal subunit | F: CTGGCACAGGCAAAGTGTCC<br>R: GCAGTGAGAGCCTTGGCTAGA | 196 | KY593120 |
| ACTB | β-actin | Cytoskeletal structural protein | F: GTCCACCGCAAGTGCTTCT<br>R: CGGTCGTGGTTGTTTCATT | 214 | KY593121 |
| TUBB | β-tubulin | Structural protein | F: GTGCTGTTGCCGATGAAAG<br>R: GCATGTCCATGAAGGAGGTT | 157 | KY593125 |
| EF1α | elongation factor 1-alpha | Essential component of the eukaryotic translational apparatus | F: GGAGATGCCAGCCTCAAAC<br>R: GATATTGCGTTGTGGAAGT | 165 | KY593123 |
| Ubiq | Ubiquitin | Protein degradation | F: GCCGAGGCTACATTCCAGT<br>R: GAAGCTTGACATGACCACGAT | 203 | MF680835 |
| 18S RNA | 18S rRNA | rRNA in the ribosome | F: TCCGCAGGAGTTGCTTCGAT<br>R: ATTAAGCCGCAGGCTCCACT | 142 | FJ843070 |

Wilmington, DE, USA) was employed to evaluate the RNA quality and concentration. RNA with a 260/280 ratio between 1.8 and 2.2 and 260/230 ratio >1 and <3 was considered satisfactory for use in experiments.

## Selection of internal control genes

Seven commonly used reference genes were identified from the transcriptome of *O. reevesii*. The following are some of the criteria we applied for selecting candidate reference genes: to minimize the effect of co-regulation, the nucleic acid sequence should encode a protein that plays various roles in cellular metabolism with different molecular functions; sequences that have previously been examined for stability in, to a certain extent, similar biological contexts; for reliability, the sequence should be tested specifically to verify the accuracy of the data (*Die et al., 2017*). The previously published candidate reference genes (*Purohit et al., 2015*; *Altmann et al., 2015*; *Etich et al., 2016*) are *cyclophilin* (*CYC*), *beta actin* (*ACTB*), *elongation factor-1 alpha* (*EF1a*), *ribosomal protein L28* (*RPL28S*), *β-tubulin* (*TUBB*), *ubiquitin* (*Ubiq*) and *18S ribosomal RNA* (*18S RNA*). To select corresponding sequences, all candidate genes selected from the transcriptome were analysed in the NCBI database using BLAST, and sequences were uploaded to the NCBI database to obtain the GenBank accession number (Table 1).

## Primer design and real-time qPCR assays

The software Primer 5.0 was used to design gene-specific primers. Optimized primer pairs were selected based on their amplification efficiencies and specificities (*D'Haene, Vandesompele & Hellemans, 2010*). The specificity of the PCR primers utilized was evaluated using the melting curve produced by the Applied Biosystems™ QuantStudio™ 6 Flex Real-Time PCR System (Thermo Fisher, Waltham, MA, USA). The fragment size of the PCR product was determined by 2.0% agarose gel electrophoresis. Primer standard curves
were created using a cDNA dilution series, and amplification efficiencies were calculated using the following equation: $E = (10^{-1/\text{slope}} - 1) * 100\%$ (*Radonić et al., 2004*). A correction for different amplification efficiencies was introduced in the sample quantification process (*Marino & Cook PMiller, 2003*).

NovoStart$^{®}$ Reverse Transcriptase (NOVOPROTEIN, Shanghai, China) was used to reverse transcribe 2000 ng of total RNA from each sample into cDNA in a 20 µl volume. To allow for increasing the volume of template cDNA for subsequent experiments, stock cDNA samples were diluted 5-fold for real-time PCR. Applied Biosystems$^{TM}$ QuantStudio$^{TM}$ 6 Flex Real-Time PCR System was employed for real-time PCR using a 20 µl reaction system containing the following components: 2 µl cDNA sample, 10 µl 2×NovoStart$^{®}$ SYBR qPCR SuperMix (NOVOPROTEIN, Shanghai, China), 0.8 µl each primer, 0.4 µl ROX Reference Dye II and 6.0 µl ddH$_2$O. Following denaturation at 94 °C for 5 min, 40 cycles of melting at 95 °C for 15 s, annealing at 57 °C for 20 s and extension at 72 °C for 30 s were carried out. A melting curve analysis was also performed.

### Data analysis

Three software tools, geNorm, BestKeeper, and NormFinder, were used to evaluate reference gene stability according to their respective (*Vandesompele et al., 2002*; *Andersen, Jensen & TF, 2004*; *Pfaffl et al., 2004*). The geNorm program ranks the most stable reference genes based on the average pairwise variation of a reference gene with other selected housekeeping genes and sorts reference genes using their expression stability value (M). In general, the lower the M value, the higher the expression stability. BestKeeper predicts "ideal" reference genes according to pair-wise correlation analysis among all pairs of candidate reference genes. NormFinder calculates the entire variation of candidate reference genes in all samples and also performs intragroup and intergroup comparisons. RefFinder (https://omictools.com/reffinder-tool) is a user-friendly web-based comprehensive tool developed for evaluating and screening reference genes from extensive experimental datasets. It integrates the currently available major computational programs (geNorm, NormFinder, BestKeeper) to compare and rank candidate reference genes. Based on the rankings from each program, RefFinder assigns an appropriate weight to an individual gene and calculates the geometric mean of their weights for the overall final ranking (*Kim et al., 2010*).

## RESULTS

### Real-time PCR amplification efficiencies

To ensure comparability among the seven housekeeping genes evaluated, PCR amplification efficiencies were calculated based on the slopes resulting from the measurement of cDNA serial dilutions. All tested reference gene amplification efficiencies were in the range of 90–110% (*RPL28S* (94.3%), *ACTB* (97.03%), *TUBB* (97.32%), *CYC* (96.61%), *EF1α* (92.03%), *18S RNA* (95.54%) and *Ubiq* (100.89%)). The melting curves for all amplification products demonstrated a single peak, verifying primer-specific amplification (Fig. S1).

**Table 2** Statistical results and expression level analyses of all tested candidate reference genes in different tissues on account of their threshold cycle point values (Ct) as furnished by BestKeeper.

| Factor | CYC | RPL28S | ACTB | TUBB | EF1a | Ubiq | 18S RNA |
|---|---|---|---|---|---|---|---|
| N | 6 | 6 | 6 | 6 | 6 | 6 | 6 |
| GM[Ct] | 28.94 | 27.76 | 21.98 | 23.05 | 19.73 | 31.07 | 22.58 |
| AM[Ct] | 28.96 | 27.77 | 22.16 | 23.06 | 19.75 | 31.09 | 22.65 |
| Min[Ct] | 27.88 | 26.88 | 18.75 | 21.74 | 19.1 | 29.96 | 20.08 |
| Max[Ct] | 30 | 28.92 | 26.91 | 24.4 | 21.35 | 33.09 | 25.18 |
| SD[ ±Ct] | 0.89 | 0.48 | 2.67 | 0.77 | 0.66 | 0.94 | 1.72 |
| CV[%Ct] | 3.09 | 1.74 | 12.05 | 3.33 | 3.33 | 3.01 | 7.58 |
| Min[x-fold] | −2.09 | −1.84 | −9.37 | −2.47 | −1.55 | −2.17 | −5.63 |
| Max[x-fold] | 2.09 | 2.23 | 30.66 | 2.56 | 3.08 | 4.06 | 6.08 |
| SD[± x-fold] | 1.86 | 1.4 | 6.37 | 1.7 | 1.58 | 1.91 | 3.29 |

## Expression stability of the tested genes

Based on the statistical results of expression analyses for each housekeeping gene in the seven different tissue types according to BestKeeper (Table 2), genes showing the lowest expression level were *CYC* and *Ubiq*, with Ct mean (AM[Ct]) values of 28.96 and 31.09 cycles, respectively. The genes exhibiting the highest expression levels were *ACTB*, *TUBB* and *EF1α*, with AM[Ct] values of 22.16, 23.06 and 19.75 cycles, respectively. *RPL28S* displayed the least variation in expression among the analysed tissues [SD (±Ct) = 0.48], whereas *18S RNA* [SD(±Ct) =1.72] and *ACTB* [SD(±Ct) = 2.67] showed the greatest variation. Candidate reference genes with an SD value higher than 1 must be considered unsuitable (*Mehta et al., 2010*), and the M value of *ACTB* according to the geNorm program was clearly higher than 1.5. Overall, the ranking of these candidate housekeeping genes by BestKeeper, NormFinder and geNorm programs was consistent. In addition, *ACTB* and *18S RNA* appeared to be less stable, whereas *RPL28S*, *EF1 α*, and *TUBB* were the most stable genes (Table 2; Fig. 1 Group1, Fig. 2A). According to geNorm, a V2/3 value of 0.15 is the proposed cut off value, under which the inclusion of an additional reference gene is not necessary (Fig. S2A). In this study, the V2/3 value was 0.265, and the values of V3/4, V4/5, V5/6, and V6/7 were all greater than 0.15, indicating no need to include another gene in as a normalization factor. Thus, the optimal number of reference genes for normalization in this example is one. The comprehensive ranking of the seven candidate genes given by RefFinder indicated *RPL28S* as the best reference gene. *EF1a* was also found to be an ideal reference gene when abundant reference gene expression is required.

## Expression stability in muscle of the seven genes under different salinity treatments

Considering that the salinity of *O. reevesii*'s habitat often fluctuates greatly due to the influence of tides, river flow, rainfall and other factors, multiple salinity gradients were established to observe changes in housekeeping gene expression to explore the response of *O. reevesii* to changes in salinity (osmotic pressure regulation ability).

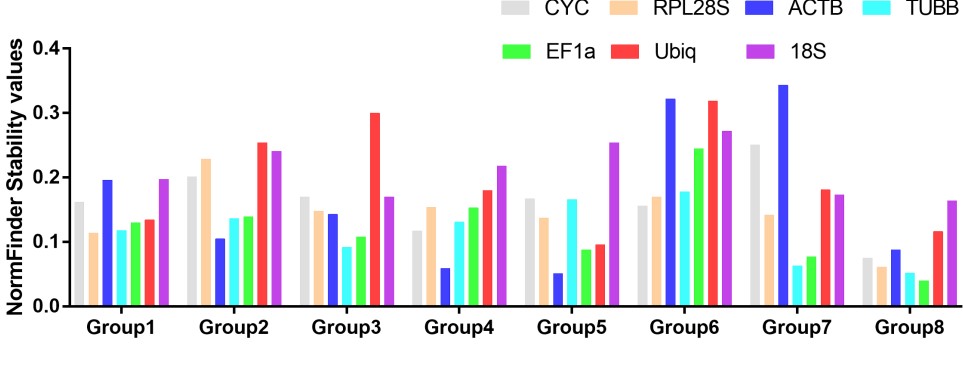

**Figure 1 Stability values of the seven housekeeping genes evaluated by NormFinder.** Different tissues (Group1); dorsal muscle tissue under 0 **ppt** salinity (Group2); dorsal muscle tissue under 5 **ppt** salinity (Group3); dorsal muscle tissue under 15 **ppt** salinity (Group4); dorsal muscle tissue under 25 **ppt** salinity (Group5); dorsal muscle tissue under 35 **ppt** salinity (Group6); dorsal muscle tissue after injury (Group7); dorsal muscle tissue from animals of different weights (Group8).

## 0 ppt salinity

Under conditions of varying salinity, BestKeeper, NormFinder, and geNorm were employed to examine the expression level of each housekeeping gene. According to BestKeeper (Table 3), the expression stability ranking of the seven reference genes for the seven sampling time points was as follows: *18S RNA >RPL28S >ACTB >TUBB >Ubiq >EF1a >CYC*. Additionally, geNorm analysis showed that *TUBB*, *EF1a* and *ACTB* had the highest stabilities (Fig. 1 Group 2). NormFinder analysis revealed *ACTB* to be the gene with the greatest stability (Fig. 2B). Because the Vn/n+1 values provided by geNorm were all greater than 0.15, multiple housekeeping genes were not required as internal references (Fig. S2B). Based on the results of a comprehensive analysis by RefFinder, we recommend that *TUBB* be used as an internal reference for qRT-PCR analyses of *O. reevesii* under this condition. However, when the reference gene should exhibit a high level of expression, *ACTB* is the most appropriate choice.

## 5 ppt salinity

According to BestKeeper (Table 4), the expression stability ranking of the seven candidate genes was *18S RNA >RPL28S >EF1a >ACTB >TUBB >Ubiq >CYC*, an order that was slightly different from the result for the 0 ppt salinity condition. However, except for *18S RNA* (0.68), the SD[±Ct] values for the housekeeping genes were greater than 1. NormFinder (Fig. 1 Group3) and geNorm (Fig. 2C) indicated *EF1a*, *RPL28S* and *TUBB* to be the most stable genes, with Vn/n+1 values all greater than 0.15 (Fig. S2). Although the results from the three software programs are inconsistent and not ideal, according to RefFinder, *RPL28S* should be used as a reference gene under this condition. Due to the insufficient level of *18S RNA*, *RPL28S* and *EF1a* expression, *ACTB* should be used as a reference gene under the stress of this level of salinity.

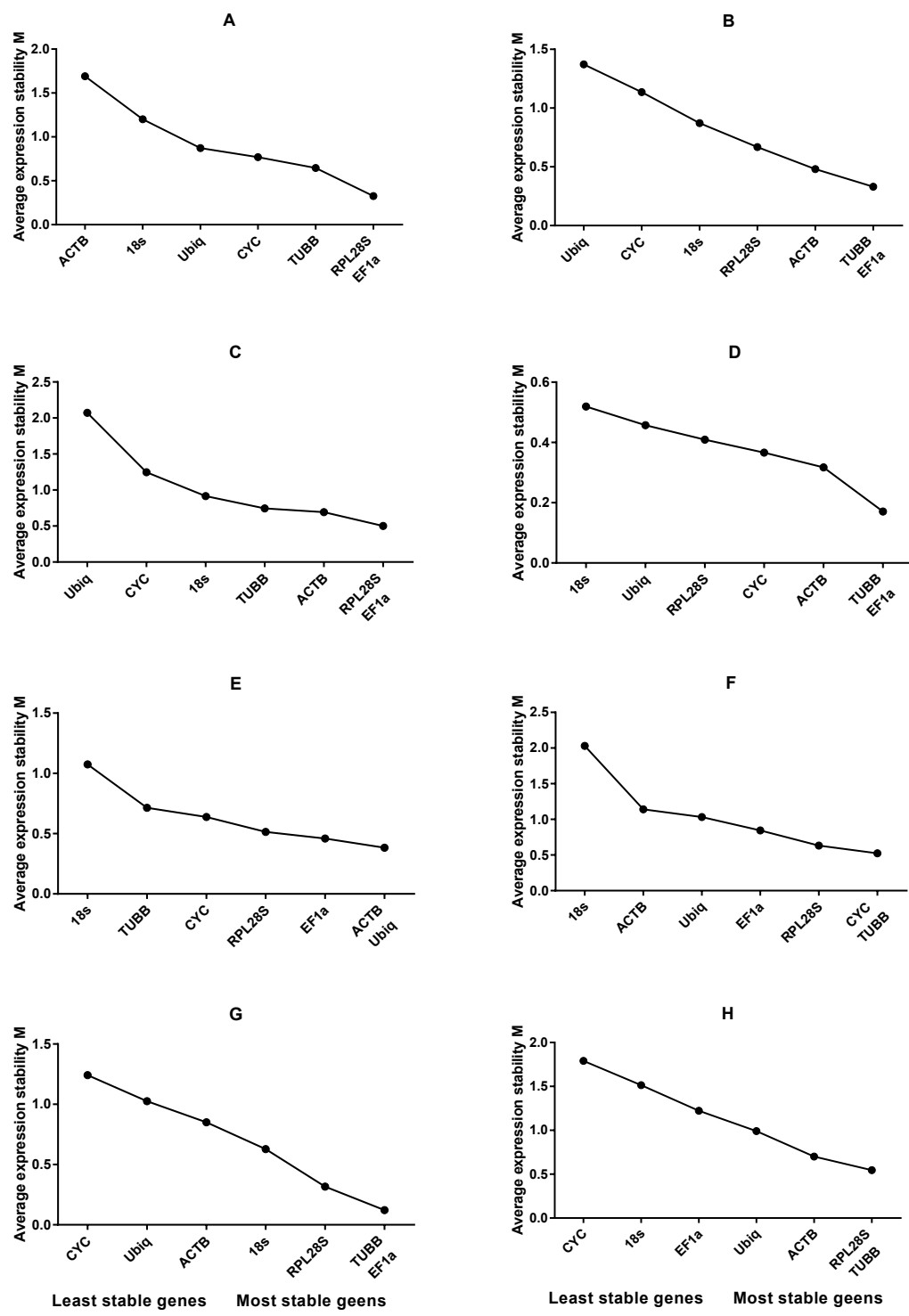

**Figure 2 Average expression stability values of housekeeping genes analysed by geNorm.** (A) Different tissues; (B) dorsal muscle tissue under **0 ppt** salinity; (C) dorsal muscle tissue under **5 ppt** salinity; (D) dorsal muscle tissue under **15 ppt** salinity; (E) dorsal muscle tissue under **25 ppt** salinity; (F) dorsal muscle tissue under **35 ppt** salinity; (G) dorsal muscle tissue after injury; (H) dorsal muscle tissue from animals of different weights.

**Table 3** Statistical results and expression level analyses of all tested candidate reference genes under culturing environment in salinity of 0ppt on account of their threshold cycle point values (Ct) as furnished by BestKeeper.

| Factor | CYC | RPL28S | ACTB | TUBB | EF1a | Ubiq | 18S RNA |
|---|---|---|---|---|---|---|---|
| N | 7 | 7 | 7 | 7 | 7 | 7 | 7 |
| GM[Ct] | 32.12 | 31.65 | 23.24 | 28.08 | 28.6 | 36.19 | 32.65 |
| AM[Ct] | 32.19 | 31.66 | 23.26 | 28.1 | 28.62 | 36.22 | 32.66 |
| Min[Ct] | 29.13 | 30.72 | 22.43 | 26.65 | 27.16 | 33.23 | 31.52 |
| Max[Ct] | 35.74 | 33 | 24.98 | 30.03 | 30.36 | 38.13 | 33.55 |
| SD[±Ct] | 1.65 | 0.69 | 0.7 | 0.95 | 1.14 | 1.08 | 0.64 |
| CV[%Ct] | 5.13 | 2.19 | 3.02 | 3.38 | 3.98 | 2.98 | 1.96 |
| Min[x-fold] | −7.98 | −1.91 | −1.75 | −2.69 | −2.71 | −7.77 | −2.19 |
| Max[x-fold] | 12.24 | 2.55 | 3.35 | 3.87 | 3.39 | 3.82 | 1.86 |
| SD[± x-fold] | 3.14 | 1.62 | 1.63 | 1.93 | 2.2 | 2.11 | 1.56 |

**Table 4** Statistical results and expression level analyses of all tested candidate reference genes under culturing environment in salinity of 5ppt based on their threshold cycle point values (Ct) as provided by BestKeeper.

| Factor | CYC | RPL28S | ACTB | TUBB | EF1a | Ubiq | 18S RNA |
|---|---|---|---|---|---|---|---|
| N | 7 | 7 | 7 | 7 | 7 | 7 | 7 |
| GM[Ct] | 33.15 | 30.59 | 23.35 | 27.22 | 28.48 | 35.1 | 32.13 |
| AM[Ct] | 33.26 | 30.63 | 23.41 | 27.29 | 28.52 | 35.18 | 32.14 |
| Min[Ct] | 28.87 | 29.51 | 21.44 | 25.37 | 26.78 | 30.01 | 31.06 |
| Max[Ct] | 38.67 | 33.94 | 27.08 | 32.12 | 32 | 37.79 | 34.17 |
| SD[±Ct] | 2.03 | 1.35 | 1.48 | 1.58 | 1.37 | 1.79 | 0.68 |
| CV[%Ct] | 6.1 | 4.4 | 6.31 | 5.8 | 4.79 | 5.09 | 2.11 |
| Min[x-fold] | −19.39 | −2.11 | −3.75 | −3.6 | −3.25 | −34.04 | −2.09 |
| Max[x-fold] | 46 | 10.21 | 13.26 | 29.88 | 11.54 | 6.45 | 4.12 |
| SD[± x-fold] | 4.08 | 2.55 | 2.78 | 3 | 2.58 | 3.46 | 1.6 |

## 15 ppt salinity

BestKeeper results (Table 5) showed a stability ranking of expression for the seven candidate genes of *CYC>ACTB >EF1a>RPL28S >TUBB >18S RNA >Ubiq*, which was significantly different from that of the 0 ppt and 5 ppt conditions, even though SD[ ±Ct] for all housekeeping genes was lower than 1. According to NormFinder (Fig. 1 Group4) and geNorm (Fig. 2D), *EF1a*, *ACTB*, *CYC* and *TUBB* were the most stable reference genes, and the V2/3 values of 0.127 suggested that the normalization factor should preferably consist of two housekeeping genes (Fig. S2D). The best combination of two genes was *TUBB* and *EF1a* (Fig. 2D). RefFinder indicated that *TUBB* was the most stable internal reference gene. *Ubiq* exhibited slightly lower expression, but *EF1a*, *ACTB* and *CYC* can all be used as internal reference genes for this treatment.

## 25 ppt salinity

For this condition, BestKeeper results (Table 6) revealed an expression stability ranking of *Ubiq >ACTB >EF1a >RPL28S >CYC >TUBB >18S RNA*; however, SD[ ±Ct] for *18S*
**Table 5  Statistical results and expression level analyses of all tested candidate reference genes under culturing environment in salinity of 15ppt on account of their threshold cycle point values (Ct) as furnished by BestKeeper.**

| Factor | CYC | RPL28S | ACTB | TUBB | EF1a | Ubiq | 18S RNA |
|---|---|---|---|---|---|---|---|
| N | 7 | 7 | 7 | 7 | 7 | 7 | 7 |
| GM[Ct] | 20.59 | 27.49 | 19.39 | 22.78 | 19.88 | 30.7 | 21.21 |
| AM[Ct] | 20.59 | 27.49 | 19.39 | 22.79 | 19.89 | 30.71 | 21.22 |
| Min[Ct] | 20.18 | 26.95 | 18.41 | 21.91 | 19.17 | 29.83 | 20.51 |
| Max[Ct] | 20.95 | 28.02 | 19.89 | 23.42 | 20.33 | 31.56 | 22.46 |
| SD[±Ct] | 0.26 | 0.34 | 0.3 | 0.36 | 0.32 | 0.58 | 0.51 |
| CV[%Ct] | 1.25 | 1.23 | 1.55 | 1.6 | 1.61 | 1.89 | 2.42 |
| Min[x-fold] | −1.33 | −1.46 | −1.98 | −1.83 | −1.64 | −1.83 | −1.63 |
| Max[x-fold] | 1.29 | 1.44 | 1.42 | 1.56 | 1.36 | 1.81 | 2.39 |
| SD[± x-fold] | 1.2 | 1.26 | 1.23 | 1.29 | 1.25 | 1.49 | 1.43 |

**Table 6  Statistical results and expression level analyses of all tested candidate reference genes under culturing environment in salinity of 25ppt on account of their threshold cycle point values (Ct) as furnished by BestKeeper.**

| Factor | CYC | RPL28S | ACTB | TUBB | EF1a | Ubiq | 18S RNA |
|---|---|---|---|---|---|---|---|
| N | 7 | 7 | 7 | 7 | 7 | 7 | 7 |
| GM[Ct] | 22.22 | 27.75 | 19.44 | 20.35 | 18.48 | 31.5 | 20.77 |
| AM[Ct] | 22.23 | 27.75 | 19.44 | 20.37 | 18.49 | 31.5 | 20.86 |
| Min[Ct] | 20.76 | 26.82 | 19.14 | 19.09 | 17.5 | 31.1 | 19.32 |
| Max[Ct] | 23.4 | 28.62 | 20.6 | 21.86 | 19.85 | 31.9 | 25.25 |
| SD[±Ct] | 0.64 | 0.58 | 0.33 | 0.87 | 0.52 | 0.22 | 1.57 |
| CV[%Ct] | 2.88 | 2.07 | 1.72 | 4.27 | 2.8 | 0.71 | 7.54 |
| Min[x-fold] | −2.74 | −1.9 | −1.23 | −2.39 | −1.97 | −1.31 | −2.72 |
| Max[x-fold] | 2.26 | 1.83 | 2.23 | 2.85 | 2.59 | 1.32 | 22.39 |
| SD[± x-fold] | 1.56 | 1.49 | 1.26 | 1.83 | 1.43 | 1.17 | 2.97 |

*RNA* was lower than 1. Different from the above conditions, the NormFinder (Fig. 1 Group5) and geNorm (Fig. 2E) results were similar to those of BestKeeper. V2/3 values were 0.156, indicating no need for multiple housekeeping genes as internal references (Fig. S2E). Although the *Ubiq* gene was found to be the most stable, its expression level was far lower than that of the *ACTB* gene. Therefore, the best reference gene is *ACTB*.

## 35 ppt salinity

*TUBB >CYC >RPL28S >Ubiq >ACTB >EF1a >18S RNA* was the expression stability ranking according to BestKeeper (Table 7). As with the condition of 25 ppt salinity, SD[±Ct] values were lower than 1, except for *18S RNA*, and the NormFinder (Fig. 1 Group6) and geNorm (Fig. 2F) results were similar to those of BestKeeper. As V2/3 values were over 0.15, multiple housekeeping genes would not be needed (Fig. S2F). The expression abundance of all candidate genes was within the acceptable range, after considering both expression stability and abundance, the best reference gene was found to be *TUBB*.

**Table 7  Statistical results and expression level analyses of all tested candidate reference genes under culturing environment in salinity of 35ppt on account of their threshold cycle point values (Ct) as furnished by BestKeeper.**

| Factor | CYC | RPL28S | ACTB | TUBB | EF1a | Ubiq | 18S RNA |
|---|---|---|---|---|---|---|---|
| N | 7 | 7 | 7 | 7 | 7 | 7 | 7 |
| GM[Ct] | 24.39 | 27.03 | 18.79 | 21.77 | 20.32 | 29.85 | 23.25 |
| AM[Ct] | 24.4 | 27.04 | 18.82 | 21.78 | 20.35 | 29.86 | 23.68 |
| Min[Ct] | 23.66 | 25.9 | 16.83 | 21.3 | 19.05 | 28.34 | 17.98 |
| Max[Ct] | 25.23 | 28.11 | 20.13 | 22.63 | 21.48 | 31.27 | 31.93 |
| SD[±Ct] | 0.44 | 0.51 | 0.77 | 0.34 | 0.89 | 0.64 | 3.79 |
| CV[%Ct] | 1.8 | 1.9 | 4.1 | 1.57 | 4.36 | 2.16 | 16 |
| Min[x-fold] | −1.66 | −2.19 | −3.9 | −1.39 | −2.41 | −2.85 | −38.68 |
| Max[x-fold] | 1.79 | 2.12 | 2.52 | 1.81 | 2.24 | 2.66 | 410.85 |
| SD[± x-fold] | 1.35 | 1.43 | 1.71 | 1.27 | 1.85 | 1.56 | 13.83 |

## Expression stability analysis of the seven genes in muscle after injury

For each of the seven housekeeping genes, transcript abundance was assessed in three independent muscle pools collected at time points ranging from 2 h to 7 days after skin damage. According to BestKeeper (Table 8), the lowest level of expression gene was displayed by *Ubiq*, with an AM[Ct] value of 29.85 cycles. In contrast, *ACTB* and *EF1α* showed the highest levels, with AM[Ct] values of 18.75 and 18.92 cycles, respectively. The BestKeeper program indicated SD(±Ct) values lower than 1 for the reference genes other than *CYC*, *Ubiq* and *18S RNA*; the geNorm M values of the seven reference genes were also lower than 1.5. BestKeeper indicated that the gene exhibiting the least variation in gene expression was *ACTB* [SD(±Ct) = 0.46]. However, NormFinder (Fig. 1 Group7) and geNorm (Fig. 2G) showed *TUBB* to be the most stable. geNorm analysis V2/3 values were 0.137, suggesting that the normalization factor should comprise additional housekeeping genes (Fig. S2G). The most suitable two-gene combination was *TUBB* plus *EF1a* (Fig. 2G). If a single gene is used as an internal reference, RefFinder indicated *EF1a* as the most appropriate choice.

## Expression stability analysis of the seven genes in individuals of different weights

Regarding groups of *O. reevesii* of different weights, BestKeeper analysis (Table 9) revealed an SD[±Ct] lower than 1 only for *CYC* gene. In contrast, NormFinder (Fig. 1 Group8) and geNorm (Fig. 2H) showed *RPL28S*, *TUBB* and *EF1a* to be significantly more stable than the other candidate genes, with Vn/n+1 values greater than the threshold of 0.15 (Fig. S2H). Based on RefFinder comprehensive analysis, *RPL28S* is more suitable as an internal reference gene when compared to the other candidates, similar to the results for the skin injury condition. Because a reference gene requires a high level of expression, *ACTB* is the most appropriate reference gene for this experimental condition.

**Table 8  Statistical results and expression level analyses of all tested candidate reference genes during skin muscle healing on account of their threshold cycle point values (Ct) as furnished by BestKeeper.**

| Factor | CYC | RPL28S | ACTB | TUBB | EF1a | Ubiq | 18S RNA |
|---|---|---|---|---|---|---|---|
| N | 7 | 7 | 7 | 7 | 7 | 7 | 7 |
| GM[Ct] | 25.26 | 27.14 | 18.75 | 22.3 | 18.92 | 29.85 | 20.99 |
| AM[Ct] | 25.33 | 27.15 | 18.76 | 22.31 | 18.94 | 29.87 | 21.03 |
| Min[Ct] | 23.28 | 26.07 | 17.38 | 21.52 | 18.21 | 27.6 | 19.03 |
| Max[Ct] | 28.32 | 28.19 | 19.67 | 23.8 | 20.32 | 31.68 | 23.12 |
| SD[±Ct] | 1.6 | 0.63 | 0.46 | 0.52 | 0.57 | 1.01 | 1.05 |
| CV[%Ct] | 6.32 | 2.33 | 2.43 | 2.31 | 3 | 3.37 | 4.98 |
| Min[x-fold] | −3.96 | −2.1 | −2.58 | −1.71 | −1.64 | −4.75 | −3.91 |
| Max[x-fold] | 8.31 | 2.07 | 1.89 | 2.83 | 2.63 | 3.56 | 4.37 |
| SD[± x-fold] | 3.03 | 1.55 | 1.37 | 1.43 | 1.48 | 2.01 | 2.07 |

**Table 9  Statistical results and expression level analyses of all tested candidate reference genes between different individual weights on account of their threshold cycle point values (Ct) as furnished by Best-Keeper.**

| Factor | CYC | RPL28S | ACTB | TUBB | EF1a | Ubiq | 18S RNA |
|---|---|---|---|---|---|---|---|
| N | 5 | 5 | 5 | 5 | 5 | 5 | 5 |
| GM[Ct] | 24.65 | 27.6 | 20.8 | 24.22 | 22.8 | 31.02 | 26.81 |
| AM[Ct] | 24.67 | 27.72 | 20.92 | 24.37 | 23.1 | 31.09 | 27.11 |
| Min[Ct] | 23.82 | 25.18 | 18.63 | 21.82 | 19.76 | 28.45 | 21.79 |
| Max[Ct] | 25.68 | 30.97 | 24.11 | 27.85 | 27.81 | 34.42 | 32.07 |
| SD[±Ct] | 0.71 | 2.54 | 2.14 | 2.65 | 3.66 | 1.88 | 3.78 |
| CV[%Ct] | 2.88 | 9.17 | 10.23 | 10.88 | 15.83 | 6.06 | 13.94 |
| Min[x-fold] | −1.78 | −5.36 | −4.52 | −5.27 | −8.27 | −5.94 | −32.48 |
| Max[x-fold] | 2.04 | 10.37 | 9.9 | 12.34 | 32.22 | 10.59 | 38.4 |
| SD[± x-fold] | 1.64 | 5.83 | 4.41 | 6.28 | 12.6 | 3.69 | 13.72 |

## DISCUSSION

In this study, seven genes, *CYC*, *RPL28S*, *ACTB*, *TUBB*, *EF1a*, *Ubiq* and *18S RNA*, were selected as candidates for reference gene screening for use in *O. reevesii*. When using a relative quantification technique to analyse mRNA expression of a target gene, the final results are more accurate and scientific when appropriate internal reference genes are employed. In general, screening of internal reference genes should to meet the following criteria: it should be widely involved in all aspects of the organism's cellular metabolism; its expression should maintain a high level and stability in a range and with low sensitivity to changes in the external environment; candidate genes should be expressed stably under different experimental conditions.

Because of advantages of high timeliness, sensitivity and convenience, qRT-PCR is often used to screen reference genes. At present, there are three programs commonly used: BestKeeper, geNorm and NormFinder. However, due to the different modes in which data are processed by the respective software, there are some inconsistencies regarding

the output. RefFinder comprehensively evaluates the results of the above three programs and provides a relatively reasonable internal control gene rank to help the experimenter ultimately determine the optimal choice.

We examined different salinity stress levels, skin muscle injury, different tissues of *O. reevesii* adults and individuals with different weights to identify reference genes. The most suitable internal reference genes in different tissues of *Haliotis rufescens* are reportedly *RPL5S* and *CYC*; the most stable in *O. reevesii* was found to be the *RPL28S* (*Sun et al., 2012*), but *EF1a* can serve as an alternative. When analysing differences in target gene expression in various tissues of *O. reevesii*, *RPL28S* should be used as the reference gene if the abundance of target gene expression is low, whereas EF1a is recommended if target gene expression is high. Previous studies have compared the most stable internal reference genes of flatworms under salinity stress, with *GM2*-activator protein *(GM2AP)* and *ACTB* indentified; these results are similar to those for *O. reevesii*, in which the most suitable internal control genes are *TUBB* and *ACTB* (*Plusquin et al., 2012*). However, our candidate gene expression analysis showed that at low salinity (0 ppt, 5 ppt) (low osmotic pressure environment) under laboratory conditions, expression of these genes was significantly lower than that at 15 ppt and 25 ppt. One reason for this may be that a low osmotic pressure environment leads to excess moisture entering tissue, decreasing cellular metabolism and eventually housekeeping gene expression (*Mizuno & Ogawa, 2011*; *Orskov, 2010*). Nonetheless, at high salinity (35 ppt), the housekeeping genes examined essentially maintained a normal level of expression, which may be due to the strong ability of *O. reevesii* to obtain water from the external environment, ensuring the stability of its internal environment. It can be inferred from the above that *O. reevesii* is not strongly tolerant to low salinity and osmotic pressure and that its optimal living salinity is approximately 15-25 ppt. Sun found the reference genes *RPL13S* and *RPL32S* to be the most stably expressed in contused rat skeletal muscle; however, *EF1a* plus *TUBB* constitutes the most suitable internal reference combination for *O. reevesii* after tissue damage (*Sun et al., 2012*). Heavy metal ion stress in organisms is also a major focus of current research, and as tidal flat inhabitants, *O. reevesii* feeds on the surface soil of these flats and may very likely serve as an indicator of heavy metal ion pollution. This is a future research direction of our laboratory (*Jáuregui et al., 2015*; *Aydın-Önen, 2016*; *Authman et al., 2015*). Although we compared data for *O. reevesii* of different sizes, because there is no consensus regarding the relationship between its growth stage and body weight, these experimental data need further confirmation. This is the first study to screen internal reference genes for *O. reevesii* under different conditions, and the results will be useful for relative quantitation of gene expression in the future.

## CONCLUSIONS

In this study, we first ascertained and evaluated the expression stability of seven housekeeping genes for qRT-PCR normalization in *O. reevesii* tissues and under conditions of salinity stress and tissue injury. (1) In our assessment of different tissues, *RPL28S* and *EF1a* were found to be the most suitable and stable internal genes among the six tissue samples tested as well as among individuals of different weights. (2) The results suggest

that *ACTB* and *TUBB* are the most stable genes, with high expression levels when assessing salinity stress. (3) Regarding muscle injury, *EF1a* is the most stable candidate gene. (4) Among all experimental groups, data analysis of two groups (15 ppt, injury) revealed *TUBB* plus *EF1a* to constitute a suitable reference combination. Based on our results, we propose that the three housekeeping genes *ACTB*, *TUBB* and *EF1a* be the first choice of reference genes for qRT-PCR. Our experimental data indicate that *O. reevesii* has low tolerance to low osmotic pressure and that a salinity range of approximately 15-25 ppt is the most suitable living environment for this organism. To our knowledge, this study is the first to select and evaluate optimal reference genes for *O. reevesii*, and the findings are expected to provide theoretical data support for future experiments involving qPCR. Although the optimal internal reference gene differs among treatments, such as during salinity stress and tissue injury, it is important to understand the importance of the selection of these genes. Overall, for different experimental studies of *O. reevesii*, the selection of appropriate reference genes should be taken into consideration, and our results provide basic experimental data for this purpose.

## ACKNOWLEDGEMENTS

We gratefully acknowledge the assistance of Mrs. Xu in the sample collection. We also thank Wang Fei, Peng Maoxiao and Chen Ya for their advice and assistance in conducting this study.

### Funding

This work was supported by the National Natural Science Foundation of China (No. 41276157). The funders had no role in study design, data collection and analysis, decision to publish, or preparation of the manuscript.

### Grant Disclosures

The following grant information was disclosed by the authors:
National Natural Science Foundation of China: 41276157.

### Competing Interests

The authors declare there are no competing interests.

### Author Contributions

- Teizhu Yang conceived and designed the experiments, performed the experiments, analyzed the data, contributed reagents/materials/analysis tools, prepared figures and/or tables, authored or reviewed drafts of the paper, approved the final draft.
- Bingning Gu conceived and designed the experiments, analyzed the data, contributed reagents/materials/analysis tools, prepared figures and/or tables, approved the final draft.
- Guolyu Xu analyzed the data, approved the final draft.
- Yanmei Shi performed the experiments, contributed reagents/materials/analysis tools, approved the final draft.

- Heding Shen analyzed the data, authored or reviewed drafts of the paper, approved the final draft.
- Rongcheng Rao performed the experiments, approved the final draft.
- Hellen Lucas Mzuka authored or reviewed drafts of the paper.

## Ethics

The following information was supplied relating to ethical approvals (i.e., approving body and any reference numbers):

Shanghai Ocean University of Leicester granted Ethical approval to carry out the study within its facilities (Approval number: Shou-DW-2019-010).

## Data Availability

The raw measurements are available in Figs. S1 and S2, and Table S1.

## Supplemental Information

Supplemental information for this article can be found online at http://dx.doi.org/10.7717/peerj.6834#supplemental-information.

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
