# Peer review of "Identification of candidate reference genes for qRT-PCR normalization studies of salinity stress and injury in Onchidium reevesii"

_PeerJ, doi:10.7717/peerj.6834_

## Round 0.1 · original submission · Major Revisions

Dear Authors,

Both reviewers agree that this manuscript can be published after some major revision. Please correct the English and pay attention in clarifying the method sections and adding additional information which is currently missing. Please explain the why you chose the salinity and injury treatments as pointed by reviewers No. 2. Also look at the comment raised by reviewer No 1 on the "take home message" and further look into the attached file.

Please answer point by point in your revised version.

·

Basic reporting

The English used in this manuscript is extremely poor and I am afraid that it requires comprehensive editing by a native English speaker.
The general structure of the paper is good and the literature references in the introduction are sufficient. However, the discussion is quite short and has a mere three references. Admittedly, I am not familiar enough with the cited literature and to what extent there are further such studies (Assessing the suitability of appropriate house-keeping genes for qPCR in general, and more specifically, under varying stress/physiological conditions), however I think this section should be far more comprehensive.

The figures and tables are good with a few minor corrections which I have noted in the attached file. All the relevant raw data appears to be available.

The article is self-contained with relevant results to hypotheses except for one issue.
In the methods section (line 38) it is stated that the animals were collected and maintained according to Shen’s method (ref. 18). This paper has an English abstract, however the main text is in Chinese. I think that in such a case the relevant methodology should be provided here in English, however I leave this matter for the editor to weigh in on.

Experimental design

Everything appears to be fine with the experimental design, research questions and methodology employed. A sufficient number of genes were assessed and all the appropriate analyses conducted.

Validity of the findings

I have one suggestion to make here. There is one important (perhaps very important) “take home message” from this paper which is not highlighted sufficiently by the authors and is of potential wide-ranging implication (outside the Onchidium community). That is, that one cannot assume that a particular “housekeeping gene” which was found to be good (i.e. stable) in one series of experiments on a particular organism will still be relevant for standardization in other experiments (and certainly not in other organisms). The notion that stress induced by extreme environmental conditions (such as salinity) may alter the expression pattern of “putative” housekeeping genes is not apparent and should be highlighted. I suspect this will make the paper more appealing to a broader audience. I am not sure but this finding may even warrant changing the title of the paper accordingly because at the moment the title speaks to a very very small niche group of interest.

Additional comments

Please see the attached file with some initial English corrections I made along the way (I am afraid that there are many more), as well as further matters which need to be clarified before publication.

Reviewer 2 ·

Basic reporting

The authors have studied seven potential housekeeping genes as a candidates foe reference genes to be used in qRT-PCR in the sea slug Onchidium struma and ranked them under salinity, injury treatments, and size (weight).
The study as a practical value in the use of O. struma as a model study animal, and therefore the work publishable. However, I found the manuscript poorly written and unsatisfactorily structured. I will not specify the grammar or spelling mistakes as they are numerous but indicate about structure:
In the abstract – not clear what is the meaning of “qRT-PCR was used to assess the expression levels of…”
Last part of the introduction should indicate the aim of the study. Instead the authors placed it in the abstract.
Line 44 – not clear what the meaning of bait.
Line 45 – What is control environmental temperature means
Line 50 – not clear what is the meaning of “After treatment”
Line 51 – not clear what is the meaning of “also tissue of dorsal skin”
Line 68-73 – paragraph is too long and should be break into several sentances
Line 125 – what is SD?
Line 221 – not clear what is the meaning of “reasonable reference genes should to be able to use for the detection of any gene of interest.”

Experimental design

It is not clear why the authors chose the salinity and injury treatments.

Shen's method (line 38) should be elaborated.

in general the M&M is not clear and should be edited and re-written.

Line 62 - SYBR qPCR SuperMix is not a solution for reverse transcription of RNA.

In figure 2, the letter for each graph should be positioned inside the graph box.

Validity of the findings

The data is robust and well analyzed.
Conclusion are stated and linked to the original research question and to the aim of the study. However it seems that the authors have not elaborated about the fact that in each treatment or treatment range, results showed a different outcomes.

Additional comments

I strongly suggest that the manuscript will be edited and proofed by an English native speaker.
In addition the Discussion and Conclusion parts should be elaborated as indicated above.

---

## Round 0.2 · Minor Revisions

Dear Authors,

Please correct and clarify those minor revision. Please also correct the Mandarin file issue, according Reviewer no#1 (and be aware of their annotated manuscript)

·

Basic reporting

The manuscript has been improved greatly from the initial submission. I have made some minor English corrections throughout the manuscript in the tracked changes file. That said, I still feel that it could be better and recommend that the authors/English editor go over it one more time before its final submission.

There is one figure in the supplement called - Leicester approval. As far as I can see this document is not referenced in the manuscript and the document is entirely in Mandarin Chinese. Please refer to this document if relevant and also I recommend that it be translated or at least have another document clarifying its purpose.

Experimental design

No comment.

Validity of the findings

No Comment.

Additional comments

I am very glad to see that this manuscript has been thoroughly revised and that overall the English is of a suitable level for publication. The various technical issues which were raised in the previous submission have also been addressed adequately.

Reviewer 2 ·

Basic reporting

Dear Yang et al.
I found the manuscript to be improved based on the manuscript writing and sentence structures. However there are some weak sentences that need to be addressed. In addition - in order to strengthen the conclusions section I suggest delivering the manuscript massages in a “bulleted list”.

Introduction
Lines 34-37
“However, there inevitably are always a number of influencing factors that affect the efficiency of these reactions, such as ….”
Please rephrase the sentence

Lines 58-60
“O. reevesii has been investigated with regard to morphological, physiological and active substance perspectives, and…”
Please rephrase the sentence

Materials and methods
Lines 76-77
“The water used in the experiment was saline, and the volume of water used ensured that all the samples were retained.”
Please rephrase the sentence

Line 115
Authors indicate that PCR products were visualized in 1% agarose gel. Such percentage is best for fragments of 500 to 10,000bp , while qRT-PCR products are smaller.
Please correct

Line 122
Correct “20-ul” to “20 ul”

Discusion
Lines 261-263
“The most suitable internal reference genes in different tissues of Haliotis rufescens are reportedly…”
Please rephrase the sentence

Lines 267-278
“Sun JH found that the reference genes RPL13S and RPL32S to be the most stably
277 expressed genes in contused rat skeletal muscle; however, EF1a plus TUBB constitutes…”
Please rephrase the sentence

Lines 280-281
“This is a research direction of our laboratory in the future41-43.”
Please rephrase the sentence

Experimental design

no comments

Validity of the findings

no comments

---

## Round 0.3 · accepted · Accept

Congratulations for your upcoming paper.

#